



# High quality organic resources are most efficient in stabilizing soil organic carbon: Evidence from four long-term experiments in Kenya.

Moritz Laub[1], Marc Corbeels[2,3], Antoine Couëdel[2], Samuel Mathu Ndungu[3], Monicah Wanjiku Mucheru-Muna[4], Daniel Mugendi[5], Magdalena Necpalova[1,6], Wycliffe Waswa[3], Marijn van de Broek[1], Bernard Vanlauwe[3], and Johan Six[1]

[1]Department of Environmental Systems Science, ETH Zurich, 8092 Zürich, Switzerland
[2]CIRAD, Avenue d'Agropolis, F-34398 Montpellier, France
[3]International Institute of Tropical Agriculture (IITA), c/o ICIPE Compound, P. O. Box 30772-00100, Nairobi, Kenya
[4]Department of Environmental Sciences and Education, Kenyatta University, P.O. Box 43844-00100, Nairobi, Kenya
[5]Department of Land and Water Management, University of Embu, P.O. Box 6-60100, Embu, Kenya
[6]University College Dublin, School of Agriculture and Food Science, Dublin, Ireland

**Correspondence:** Moritz Laub (moritz.laub@usys.ethz.ch)

**Abstract.**

In sub-Saharan Africa, long-term maize cropping with low external inputs has been associated with the loss of soil fertility. While adding high-quality organic resources combined with mineral fertilizer has been proposed to counteract this fertility loss, the long-term effectiveness and interactions with site properties still require more understanding. This study used repeated

measurements over time to assess the effect of different quantities and qualities of organic resource addition combined with mineral N on the change of soil organic carbon concentrations (SOC) over time (and SOC stocks in the year 2021) in four ongoing long-term trials in Kenya. These trials were established with identical treatments in moist to dry climates, on coarse to clayey soil textures, and have been managed for at least 16 years. They received organic resources in quantities equivalent to 1.2 and 4 t C ha$^{-1}$ per year in the form of *Tithonia diversifolia* (high quality, fast turnover), *Calliandra calothyrsus* (high quality,

intermediate turnover), *Zea mays* stover (low quality, fast turnover), sawdust (low quality, slow turnover) and local farmyard manure (variable quality, intermediate turnover). Furthermore, the addition or absence of 240 kg N ha$^{-1}$ per year as mineral N fertilizer was the split-plot treatment. At all sites, a loss of SOC, rather than gain, was predominantly observed due to a recent conversion from permanent vegetation to agriculture. The average reduction of SOC concentration over 19 years in the 0 to 15 cm depth ranged from 42% to 13% of the initial SOC concentration for the control and the farmyard manure treatments at 4 t

C ha$^{-1}$ yr$^{-1}$, respectively. Adding *Calliandra* or *Tithonia* at 4 t C ha$^{-1}$ yr$^{-1}$ limited the loss of SOC concentrations to about 24% of initial SOC, while the addition of saw dust, maize stover (in 3 of 4 sites) and sole mineral N addition, showed no significant reduction in SOC loss over the control. Site specific analyses, however, did show, that at the site with the lowest initial SOC concentration (about 6 g kg$^{-1}$), the addition of 4 t C ha$^{-1}$ yr$^{-1}$ farmyard manure or *Calliandra* plus mineral N led to a gain in SOC concentrations. All other sites lost SOC in all treatments, albeit at site specific rates. While subsoil SOC stocks in 2021

were little affected by organic resource additions (no difference in 3 of 4 sites), the topsoil SOC stocks corroborated the results



for SOC concentrations. The relative annual change of SOC concentrations showed a higher site specificity in high-quality organic resource treatments than in the control, suggesting that the drivers of site specificity in SOC buildup (mineralogy, climate) need to be better understood for effective targeting of organic resources. Even though farmyard manure showed the most potential for reducing SOC loss, our results clearly show that maintaining SOC with external inputs only is not possible at
organic resource rates that are realistic for small scale farmers. Thus, additional agronomic interventions such as intercropping, crop rotations or strong rooting crops may be necessary to maintain or increase SOC.

## 1  Introduction

Maize cropping in Sub-Saharan Africa (SSA) is generally characterized by yields $< 2$ t ha$^{-1}$, far below the world average of more than 5.5 t ha$^{-1}$ (FAO, 2021). This has to a large extent been attributed to a lack of nutrient inputs into these cropping systems,
leading to nutrient mining and a decrease in soil fertility, including soil organic matter loss (Vanlauwe and Giller, 2006). The longer low-input cropping systems are maintained, the stronger the soil fertility decline and the lower the ability of soils to provide nutrients to crops through the mineralization of soil organic matter (Vanlauwe et al., 2015). To counteract this soil fertility decline, sustainable intensification practices, which allow for increased crop yields while simultaneously maintaining or preferably increasing soil fertility (Pretty and Bharucha, 2014), are needed. In the context of smallholder systems in SSA,
integrated soil fertility management (ISFM) is expected to deliver on both aspects (Gentile et al., 2009; Gram et al., 2020). ISFM implies the use of improved germplasm and mineral fertilizer, which mainly target short-term productivity, combined with the application of organic resources to the soil (Vanlauwe et al., 2010), which target long-term system sustainability by providing the needed C and N inputs (Kong et al., 2005) to replenish soil organic carbon (SOC).

   Results from several experiments (Adams et al., 2020; Laub et al., 2022) and a recent meta analysis (Fujisaki et al., 2018)
suggest that, given the right quantity and quality of inputs, management and surrounding soil conditions, it is possible to increase the SOC contents in tropical soils. However, in other experiments in tropical SSA it was not even possible to maintain the SOC concentrations in the long term, even for high-input treatments (Kihara et al., 2020). This has been hypothesized to be a result of initially high SOC levels at the sites, favorable conditions for SOC decomposition, and a reduced stabilization capability of the 1:1 kaolinite clay minerals in tropical soils (Six et al., 2002; Sommer et al., 2018). Vanlauwe et al. (2015)
suggested that differences in local soil conditions may be responsible for observed differences in the success of SOC building practices and hence local adaptation to different soil conditions is needed. Land cover history also explains site specific SOC dynamics and can help to define the local potential for SOC sequestration. Indeed soils cultivated as pasture or forest have initially high SOC stocks that are difficult to maintain under cultivation (Lal, 2004).

   Besides the effect of soil geochemistry (Doetterl et al., 2015), the quality of organic resources can play an important role in
SOC formation (Puttaso et al., 2013), but their relative contribution to SOC storage is still insufficiently understood, especially in heavily weathered tropical soils. It has been established that the quality of organic resources, through the effect of litter stoichiometry (Sinsabaugh et al., 2013), initially determines the maximum potential to built new stable microbially derived SOC (Kallenbach et al., 2016). However, it is less certain if the efficiency of microbial processing of low-quality organic





resources (high C/N) can somehow be enhanced, for example by increasing the availability of nutrients from mineral sources. The latter has been suggested by comparing microbial efficiencies at low and high atmospheric N addition rates in forests (Li et al., 2021a), but has to date not been studied in agroecosystems. A recent study suggested that the combination of low and high-quality organic resources can enhance the overall microbial carbon use efficiency (Pingthaisong and Vityakon, 2021), yet we are not aware of any study that tested whether the same was possible when adding mineral mineral N to low-quality organic resources. Also, the interactions of organic resource quality with geochemistry have been understudied. While a recent lab study indicated that geochemistry may be as important as organic resource quality in the formation of new, stable SOC (Bucka et al., 2021), both need to be studied under field conditions. Thus, we are only beginning to understand the fundamental principles behind their interactions and a better understanding of all factors that regulate the microbial processing, stabilization and decomposition of SOC in soils is needed as old paradigms get cast into doubt (Cotrufo et al., 2021). For example, the idea that low-quality organic resources, rich in recalcitrant lignin and polyphenol, lead to more soil organic matter formation than high-quality resources Palm et al. (2001a, b) has been replaced by concepts that consider SOC of microbial origin, which favorably forms from high-quality resources, to be the most stable (Cotrufo et al., 2013; Denef et al., 2009). Also, the concept that soil texture plays the dominant role in determining how much SOC can be stabilized in a soil (Hassink, 1997) did not hold for soils at a local scale, differing in texture but with the same mineralogy (Schweizer et al., 2021); the latter suggests that the actual driver behind correlations of soil texture with SOC storage across large scales may be the different mineralogy and not the grain size.

As the main long-term goal of ISFM is to increase short-and long-term soil fertility, in particular SOC, there is a need to better understand the effect of the rate and quality of organic resource additions on SOC dynamics under different pedo-climatic conditions. This can help to answer the question whether organic resource quality, quantity or site properties are most important in building SOC. The answer to this question is crucial for site-specific recommendations. Yet, few studies reported on SOC dynamics following organic resource additions over time spans of decades or longer. Besides, it is not clear in how far the combined application of organic and mineral N fertilizer affects SOC dynamics. Therefore, we analyzed data from four long-term experiments conducted at four sites in central and western Kenya (established in 2002 and 2005, respectively). All four experiments had the same treatments with organic resource additions of the same quality at each site. The aims were to study (i) the efficiency of SOC build-up under the addition of organic resources with different qualities combined with or without mineral N fertilizer across sites as well as (ii) how the efficiency of SOC formation differs between sites. To guide our research, the following hypotheses were formulated:

1. Addition of high-quality organic resources (low C/N and lignin/N ratios), which are most efficiently processed by soil microbes (Cotrufo et al., 2013), leads to an increase in SOC concentration. In contrast, low-quality resources do not maintain SOC concentrations in the long term.

2. Due to a C/N ratio that is too high compared to typical microbial C/N rations, nitrogen availability is a major limitation for SOC build-up from low-quality organic resources. Therefore, low-quality resources have a higher SOC build-up if mineral N fertilizer is added compared to when no mineral N is applied.



3. The efficiency with which new SOC is formed is influenced by both the site conditions and the organic resource quality, but the resource quality plays the strongest role.

## 2 Material and methods

### 2.1 Site characteristics

**Table 1.** Locations, soil properties and climatic conditions of the study sites. Soil properties are given for the 0 - 15 cm depth layer, and are based on a measurement before experiment start (1 reference profile per site). Coordinates are given in the WGS 84 reference system. The site names in the table contain a hyperlink to display the location on google maps.

| Soil characteristics | Embu | Machanga | Sidada | Aludeka |
|---|---|---|---|---|
| Latitude | -0.517 | -0.793 | 0.143 | 0.574 |
| Longitude | 37.459 | 37.664 | 34.422 | 34.191 |
| Initial soil C (g kg$^{-1}$)[+] | 29 | 3 | 15 | 8 |
| Initial N (g kg$^{-1}$)[+] | 3.0 | 0.2 | 1.2 | 0.8 |
| Initial bulk density (g cm$^{-3}$) | 1.26 | 1.51 | 1.3 | 1.45 |
| pH (H$_2$O) | 5.43 | 5.27 | 5.4 | 5.49 |
| Clay (g kg$^{-1}$) | 598 | 132 | 557 | 134 |
| Soil type (IUSS Working Group, 2014) | Humic Nitisol | Ferric Alisol | Humic Ferralsol | Haplic Acrisol |
| Altitude (m) | 1380 | 1022 | 1420 | 1180 |
| Mean annual rainfall (mm)[*] | 1175 | 795 | 1730 | 1660 |
| Mean annual temperature (°C)[*] | 20.1 | 23.7 | 22.6 | 24.4 |
| Months of long rainy season | 3 - 8/9 | 3 - 8 | 3 - 8 | 3 - 8 |
| Months of short rainy season | 10 - 1/2 | 10 - 1/2 | 9 - 1 | 9 - 1 |

[+]By dry combustion (CHN628, LECO Corporation, Michigan, USA) [*]Means calculated based on measured data from 2005 to 2020

This study uses the combined data from four equally designed long-term experiments, located in central and western Kenya (see Table 1). The two experiments in central Kenya (i.e., in Embu and Machanga) were initiated in 2002, whilst the experiments in western Kenya (i.e., in Sidada and Aludeka) were started in 2005. All sites were under continuous maize cropping with two growing seasons per year. The long rainy season lasted usually from March until August/September and the short rainy season from September/October until January/February (Table 1). The sites were specifically selected to represent different altitudes, levels of precipitation, temperatures and soil conditions. With the sites in Sidada (1730 mm; 22.6°C) and Aludeka (1660 mm; 24.4°C) having the longest rainy seasons, the highest amounts of rainfall, and intermediate mean annual temperatures; thus representing the most favourable climate for maize production. The Embu site (1175 mm; 20.1°C) is slightly less favorable while Machanga (795 mm; 23.7°C) represents a dryer climate, where maize is at considerable risk of crop failure. The soils in Machanga and Aludeka are coarse-textured and have less than 15% of clay, while those in Sidada and Embu both contain more than 55% of clay The soils at all four sites are heavily weathered, which was also reflected in the low pH values of between





5.3 and 5.5 at the start of the experiments. The soil in Embu is a Humic Nitisol, which has more weatherable minerals than the Humic Ferralsol in Sidada, a soil that is dominated by low activity clays as well as iron and aluminium oxides. The Haplic Acrisol in Aludeka and the Ferric Alisol in Machanga are both characterized by illuvial clay accumulation in the subsoil and low base saturation. The clay minerals in the Acrisol are characterized by lower activity than those in the the Alisol (FAO, 1998; IUSS Working Group, 2014). The land-use history prior to the establishment of the experiments differed between sites. The sites in Sidada and Aludeka were savanna-type ecosystems, with shrubs, tree and witchgrass *Elymus repens*, having a tree cover of about 50%, and had been converted to low-intensity shifting cultivation approximately 25 years prior to the start of the experiments. The Embu site was initially a tropical evergreen forest, that was lost about 50 to 100 years ago, when low-intensity shifting cultivation had started. The site at Machanga was a steppe grassland, with a tree cover of around 20 to 30 %, that was only converted upon the start of the experiment. However, the area around Machanga has been subject to intensive grazing by cattle, so the site was far from undisturbed at the start of the experiment.

## 2.2 Experiment description

**Table 2.** Dry matter based mean measured chemical characteristics (and 95% confidence intervals) of organic resources applied at all sites. Measurements were available from Embu and Machanga from 2002 to 2004, all sites from 2005 to 2007 and in 2018. Significant differences in residue properties were found between the different organic resources, but not between sites and years. Same letters within the same row indicate the absence of significant differences for that property (p < 0.05). Abbreviations: n.c. = not classified.

| Measured property | *Tithonia* | *Calliandra* | Maize stover | Sawdust | Farmyard manure |
|---|---|---|---|---|---|
| Abbreviation[x] | TD1.2 (TD4)±N | CC1.2 (CC4)±N | MS1.2 (MS4)±N | SD1.2 (SD4)±N | FYM1.2 (FYM4)±N |
| C (g kg$^{-1}$)[+] | 345[b] (333-357) | 396[c] (383-409) | 397[c] (386-408) | 433[d] (416-449) | 234[a] (213-255) |
| N (g kg$^{-1}$)[+] | 33.2[d] (28.9-38.2) | 32.5[d] (28.3-37.3) | 7.2[b] (6.5-8) | 2.5[a] (2.1-2.8) | 18.1[c] (15-21.8) |
| C/N ratio | 12.4[a] (10.8-14.1) | 13.6[a] (11.9-15.5) | 58.7[b] (52.8-65.2) | 199.1[c] (174.1-227.7) | 12.3[a] (9.9-15.4) |
| P (g kg$^{-1}$)[#] | 2.3[d] (1.8-2.9) | 1.1[c] (0.8-1.5) | 0.4[b] (0.3-0.6) | 0.1[a] (0-0.2) | 3.1[d] (2.3-3.9) |
| K (g kg$^{-1}$)[#] | 37.2[c] (21.2-65.2) | 8.7[b] (5-15.3) | 9[b] (6-13.5) | 2.8[a] (1.6-4.9) | 19.4[bc] (7.8-48.6) |
| Lignin (g kg$^{-1}$)[#] | 90[ab] (62-117) | 105[b] (77-133) | 48[a] (37-60) | 172[c] (144-199) | 198[c] (154-242) |
| Polyphenols (g kg$^{-1}$)[#] | 19[c] (14.9-24.3) | 108.7[d] (85.3-138.6) | 11.3[b] (9.5-13.6) | 4.9[a] (3.8-6.2) | 7.8[ab] (5.2-11.5) |
| Ligin/N ratio | 2.6[a] (1.8-3.7) | 3.1[ab] (2.2-4.3) | 6.2[c] (4.8-8) | 58.3[d] (41.1-82.8) | 6.9[bc] (3.9-12.3) |
| Quality / turnover speed* | High / fast | High / intermediate | Low / fast | Low / slow | n.c. |
| Class* | 1 | 2 | 3 | 4 | n.c. |
| kg N in 4.0 t C ha$^{-1}$ yr$^{-1}$, -N [+N] | 323 [563] | 295 [535] | 68 [308] | 20 [260] | 324 [564] |
| kg N in 1.2 t C ha$^{-1}$ yr$^{-1}$, -N [+N] | 97 [337] | 88 [328] | 20 [260] | 6 [246] | 97 [337] |

[x]For 1.2 (or 4.0) t C ha$^{-1}$ yr$^{-1}$ treatments; [+]By dry combustion (CHN628, LECO Corporation, Michigan, USA); [#]Total digestion (P, K), acid detergent fibre (lignin) and Folin-Denis (polyphenols) according to Anderson and Ingram (1993); * according to Palm et al. (2001a); +N and -N indicate 120 or 0 kg mineral N fertilizer application per growing season.





All four experiments were conducted by using a split plot design with three replicates. The main treatments consisted of the addition of five types of organic resources, applied in quantities of 1200 and 4000 kg C ha$^{-1}$ yr$^{-1}$ on plots of 12x5 m (Embu) or 12x6 m (other sites). The subplot treatment consisted of the application of 120 kg mineral N ha$^{-1}$ season$^{-1}$ (+N treatment) compared to a no N input (-N) treatment. Mineral N (CaNH$_4$NO$_3$) was applied twice during each growing season: the first 40 kg N ha$^{-1}$ at planting and the remaining 80 kg N ha$^{-1}$ about six weeks later, as top dressing. In each growing season, all

plots received a blanket application of 60 kg P ha$^{-1}$ as triplesuperphosphate and 60 kg K ha$^{-1}$ as muriate of potash at planting. The organic resources were applied once a year just before planting at the start of the long rainy season. The incorporation was done using a hand hoe to a soil depth of about 15 cm. The applied organic resources represented all four quality classes that were defined by Palm et al. (2001a), differing in N, lignin and polyphenol contents (Table 2): pruned leaves including stems of <2cm thickness from *Tithonia diversifolia* (TD; high quality and fast turnover; class 1), pruned leaves including small

stems from *Calliandra calothyrsus* (CC; high quality and intermediate turnover; class 2), stover of *Zea mays* (MS; low quality and fast turnover; class 3), sawdust from *Grevillea robusta* trees (SD; low quality and slow turnover; class 4) and locally available farmyard manure (FYM; no defined class, but considered of intermediate to high quality with intermediate turnover; Sileshi et al., 2019; Silva et al., 2014). A treatment without any organic resource addition served as the control (CT). All maize residues were removed from the plots at harvests, so the only C inputs from the maize crop were the roots (and root

exudates). In addition, a randomly allocated quarter of each split plot was kept as bare fallow throughout the entire duration of the experiment, with no maize planted and with all emerging weeds removed by regular weeding. This was done to study the SOC dynamics without any additional inputs from roots or other plant debris. Further details on agricultural management can be found in Chivenge et al. (2009), Gentile et al. (2011) and Laub et al., (in review).

### 2.2.1   Soil sampling

We used pooled data from all soil sampling campaigns that were conducted throughout the experiments, the latest sampling being in 2021. In Embu and Machanga, regular soil sampling campaigns were done every two to three years since the first experimental year (2002), while in Sidada and Aludeka, regular soil samplings were only initiated in 2018, due to budget constraints. With the exception of the 2021 sampling, only topsoil (0-15 cm) samples from the cropped plots were taken, by combining a composite sample of six transect insertions along the two diagonals. During the 2021 sampling campaign, soil

samples were taken down to 50 cm depth (the depth intervals were 0-15-30-50 cm) and the bare plots were also sampled. The initial aim was to sample even deeper, but due the soil being too hard and crumbly (Embu) or too shallow (Aludeka), it turned out unfeasible to sample from deeper layers. To minimize plot disturbance, especially below ploughing depth, the 2021 samples were collected by only one soil core from the center of each plot, using a gauge auger of 60 mm diameter and 500 mm length in Embu (Eikelkamp Soil & Water, Giesbeek, Netherlands) and a soil corer of 55 mm diameter and 1000 mm length

(Giddings Machine Company, Windsor, USA) in all other sites. This allowed for bulk density (BD) estimations by weighing fresh samples, determining the soil moisture content based on a sub-sample and subsequently computing the dry soil weight per known core volume. All soil samples were sieved through an 8 mm sieve immediately after sampling and then air dried for storage until further analysis in the laboratory. Samples were then broken and crushed by pestle and mortar and sieved through





a 2 mm sieve. Prior to analysis, samples were finely ground with a ball mill, then soil C and N concentrations were measured

by dry combustion using an elemental analyzer (CHN628, LECO Corporation, Michigan, USA). In addition, soil pH ($H_2O$)
was determined on the 0-15 cm topsoil samples taken in 2018. Because soil pH values lower than 6.5 were observed at all sites,
no correction for carbonates was necessary in the calculation of SOC concentrations.

### 2.3    Calculation of SOC stocks based on equivalent soil mass

As the measurements conducted in 2021 consisted of both SOC and BD, the equivalent soil mass approach was used to estimate

SOC stocks across normalized soil masses (Lee et al., 2009), in addition to SOC concentrations. For this, the approach of Wendt
and Hauser (2013) was followed, by fitting a cubic spline to measured data in order to re-scale to equivalent soil masses. First,
the soil masses and SOC stocks were computed for each soil depth layer that was sampled. Then, cumulative SOC stocks and
soil masses were calculated for combined depth layers to increasing depth (0-15, 0-30 and 0-50 cm) by summing the SOC
stocks and soil masses of the depth layer with the values from all above layers. Next, a cubic spline with no intercept was fitted

to each individual sampled soil core with cumulative SOC stocks as dependent and cumulative soil masses and the squared
value thereof as independent variables. From visual inspection of this fit and $R^2$ values of 0.99, it was concluded that the
fitted cubic splines could be used to compute SOC stocks for normalized cumulative soil masses for each sampled soil core.
We chose normalized cumulative soil masses of 0-2500 t soil ha$^{-1}$ and 2500-7500 t soil ha$^{-1}$, which across the sites roughly
corresponded to the mean soil masses in the depths of 0-15 cm and 15-50 cm, respectively.

### 2.4    Statistical analysis

#### 2.4.1    Creation of statistical models

Statistical analyses of SOC and soil total N concentrations in 0 to 15 cm depth and their temporal trends were performed using
mixed linear models. Random slopes and intercepts of the temporal effect were initially included in a nested structure down to
the split plot level. Because of an almost perfect correlation between changes in SOC and soil total N concentrations (Fig. A1),

the focus of this paper was on SOC concentrations. Two different statistical models were developed. One model which focused
on site-specific effects, and another model that focused on the general trend of SOC concentrations across sites, in which site
was a random effect. For the latter model, the SOC data was normalized to the percentage of initial SOC, i.e., data points from
each site were divided by the mean SOC concentration of each experimental replicate (block) at the start of the experiment
(as initial measurements were not available for all plots). The initial fixed effects in the model were interactions of time since

the experiment started with the organic resource treatment, the mineral N treatment and the interaction of organic resource
treatment with mineral N treatment. In the site-specific model, all these fixed effects were further allowed to be site specific,
by adding an interaction with site for each of them. In this model, site was the only fixed effect that was also allowed to have
an intercept of its own and not only an interaction with time. This assured that different treatments at the same site would start
at the same SOC concentration at the start of the experiment in the site-specific model, and that all treatments and sites would

start at initially 100% of SOC in the model that treated site as a random effect. In both statistical models, random effects were





systematically eliminated one by one until the best random effects structure with the lowest Akaike Information Criterion (AIC) was found. This resulted in the removal of the random slope in the models but a random intercept was kept. The nested random intercepts were dataset/blockInSite/Plot and dataset/Site/blockInSite/Plot, for the models without and with site as random effect, respectively. The random effect of "dataset" accounted for potential systematic effects of the soil sampling in different

years. After selecting an appropriate random effects structure and visually checking for normality and homoscedasticity (a site-specific variance had to be included), interactions of fixed effects were removed one by one until only significant fixed effects remained, starting at the highest order of interaction. Selection of random effects was done using restricted maximum likelihood, that of fixed effects with maximum likelihood and the final model was then fit again with restricted maximum likelihood (Zuur et al., 2009). Additionally, an alternative version of the site-specific model, using relative SOC data (percent

of initial SOC) as dependent variable, was created to compare SOC changes within the same organic resource treatments across sites.

All statistical analyses of the SOC stock data from 2021, based on equivalent soil mass, and pH data from 2018, were done in a similar way as the analyses of SOC concentration. Fixed effects were the organic resource and mineral N treatments, the site and the cropping status (cropped vs bare; in the case of SOC stocks data only) and interactions of all were allowed. Models

contained a random nested blockInSite/Plot effect and allowed for variance heterogeneity between sites, while no random effect for "dataset" and no interaction with time was needed, given the single time point.

### 2.4.2 Estimation of carbon storage efficiency

From the temporal trends of SOC concentration, we further estimated the change of SOC stocks in 0-15 cm depth, with the goal to derive the apparent carbon storage efficiency ($CSE_a$) of the different organic resources, i.e., a measure of efficiency to

retain C. The $CSE_a$ has been defined as the fraction of C inputs contributing to C storage in the soil (Manzoni et al., 2018), e.g., in our case how much the annually added C through organic resources is found in the soil compared to the control treatment. To do so, we multiplied the least square means of the change in annual SOC concentration obtained by site and treatment from the mixed model, with the mean BD of each site estimated from topsoil measurements to obtain the mean annual change in SOC stocks (treatment-specific differences in BD were absent). These BD estimations were also derived using a mixed linear

model, from the available BD measurements which had been conducted in the experimental year 1, 2 and in the calendar year 2021 at each of the sites. Then, a site- and organic resource-specific linear regression between the estimated annual change in SOC stocks and the amount of annual organic resource C applied was fit, using a site-specific intercept. The slope of this regression was taken as an estimate for $CSE_a$ of the different organic resources at the different sites (Manzoni et al., 2018) and we tested whether significant differences existed in the $CSE_a$ between the different organic resources at the different sites (i.e.,

testing for a significant effect of organic resource treatment, site and their interaction).

### 2.4.3 Statistical software and definitions

The R software version 4.0.4 (R Core Team, 2021) with the 'nlme' package (Pinheiro et al., 2021) was used for all statistical analyses. The post-hoc pairwise comparison of different treatments at different times was done with the 'cld' function (Piepho,





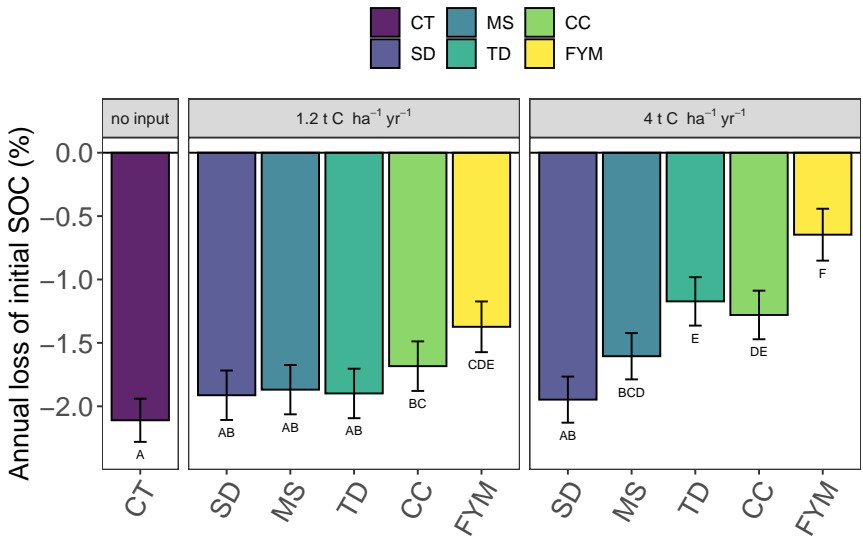

**Figure 1.** Annual changes of SOC concentrations in the top 0-15 cm soil layer in different organic resource treatments across sites. Annual change in SOC concentrations is displayed as percentage of initial SOC. Treatments that share capital letters do not differ significantly from each other in the annual change of SOC (all p < 0.05). The error bars indicate the 95% confidence interval for the annual change of SOC. *Abbreviations: CT, control; SD, saw dust; MS, maize stover; TD, Tithonia; CC, Calliandra; FYM, farmyard manure.*

2004) of package 'emmeans' (Lenth, 2021) using the 'containment' method to estimate degrees of freedom. Estimated least
square means were computed for SOC at different time points, for the temporal trends of SOC concentrations and for SOC
stocks from 2021 and pH data at the time of measurement. Note that in the flow text, the term significant refers to the p < 0.05
threshold, if not specified otherwise.

## 3   Results

### 3.1   General trends of SOC concentrations across sites

The analysis of relative SOC change across all sites showed that even at high amounts of added organic resources, a decrease
of topsoil SOC concentration occurred over time with continuous maize cropping (Fig. 1 and 2). Significant differences in the
magnitude of annual SOC decrease existed between the different organic resource classes applied at the high rate of 4 t C ha$^{-1}$
yr$^{-1}$ and to a lesser extent at the lower rate of 1.2 t C ha$^{-1}$ yr$^{-1}$. In contrast, the application of mineral N did not have a significant
effect on annual decrease of SOC concentration (Fig. 2), so +N and -N were evaluated together. Across the four sites, SOC
concentration decreased strongest under the CT treatment. On average by 2.1% of initial SOC per year, corresponding to 40%
loss over 19 years. Average SOC decreases in all the 4 t C ha$^{-1}$ yr$^{-1}$ treatments, except SD, were significantly lower than in
the control but did not completely halt SOC losses. The FYM and TD treatments resulted in the lowest decrease of SOC



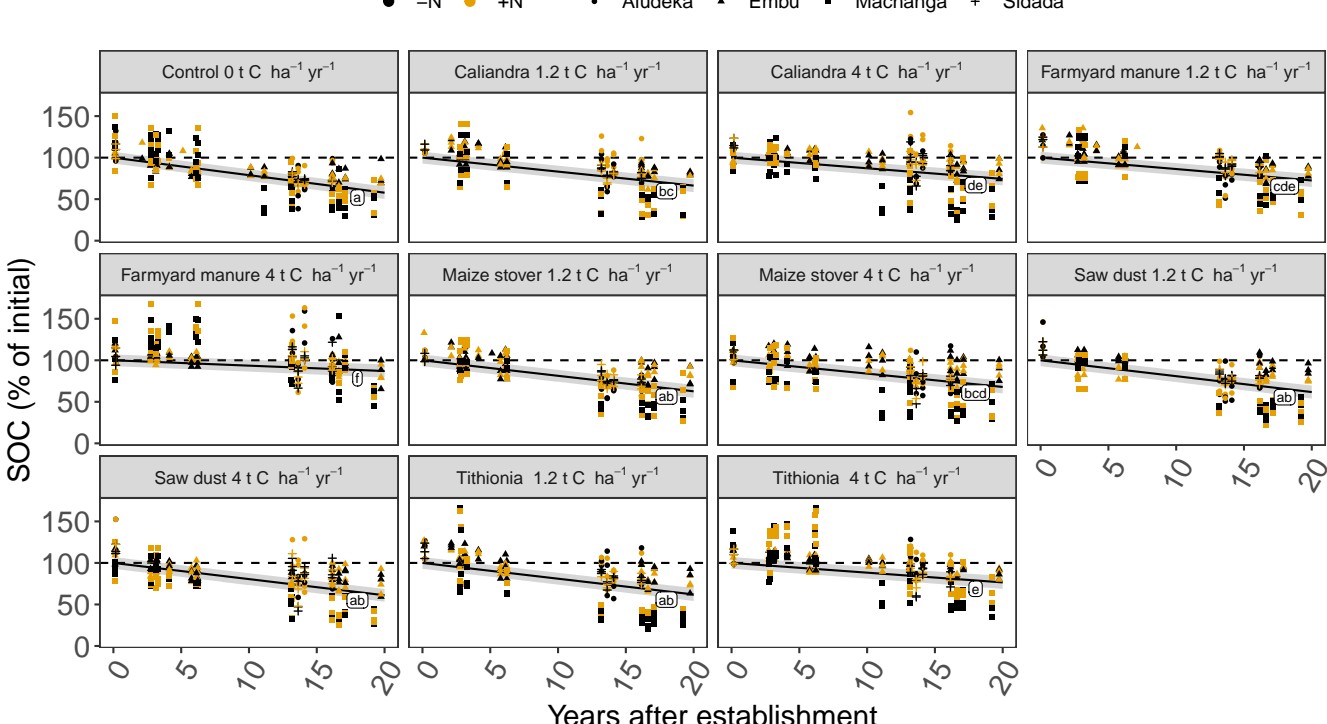

**Figure 2.** General trend in the SOC concentrations in the top 15 cm soil layer under different organic resource treatments across sites. The change in SOC concentrations is shown as a percentage of initial SOC concentrations. Same lowercase letters indicate the absence of a significant difference in the temporal development between different organic resource treatments, while the mineral N additions did not have a significant effect (all $p < 0.05$). Large initial fluctuations in the Machanga site are a combined result of measurement variability and low SOC contents. The grey shaded areas constrained by the dashed lines indicate the 95% confidence intervals for true mean of SOC at different times.

concentration, i.e., 0.6 and 1.2% of initial SOC per year, respectively (12 and 23% over 19 years). Notably, the FYM treatment at 4 t C ha$^{-1}$ yr$^{-1}$ resulted in significantly lower SOC decrease than all other treatments, followed by TD at 4 t C ha$^{-1}$ yr$^{-1}$, which

was significantly lower than all but the CC treatment at 4 t C ha$^{-1}$ yr$^{-1}$. The application rates of 1.2 t C ha$^{-1}$ yr$^{-1}$ were much less effective in reducing SOC decrease; only FYM and CC were significantly different from the control, with average decreases of 1.4 and 1.7% of initial SOC per year, respectively.

### 3.2   Site specific differences in the influence of organic resource quality on SOC development

The analysis of SOC concentration at the site level revealed strong site specificity of temporal changes in relation to the different

organic resource treatments. For example in Aludeka, the FYM treatment at 4 t C ha$^{-1}$ yr$^{-1}$ showed a significant gain in SOC of 0.1 g C kg$^{-1}$ yr$^{-1}$, while in Embu, all treatments lost at least 0.4 g C kg$^{-1}$ yr$^{-1}$ (Fig. 4). The order of treatments, however, was





**Figure 3.** Temporal trends of SOC concentrations in the top 15 cm of the soil by site, organic resource and N treatment, displayed in combination with measured raw data, shown by the crosses. Same lowercase letters indicate the absence of significantly different temporal trends between treatments (all p < 0.05). The grey shaded areas constrained by the dashed lines indicate the 95% confidence intervals for the true mean of SOC at different times.





**Figure 4.** Temporal trend of SOC concentrations in the top 15 cm of the soil by site, organic resource and N treatment in both absolute and relative values. Bars in lighter colors with grey outlines represent the -N treatments and bars in darker colors with black outlines represent the +N treatment. Statistics across sites were done based on the relative SOC changes (all p < 0.05). Same lowercase letters indicate the absence of a significant difference in the temporal trend between treatments at the same site. Same uppercase letters indicate the absence of a significant difference within the same organic resource treatment across sites (resource type and amount, including the comparison between +N and -N of the same treatment). As a reference to the general trends, the grey dashed line indicates the mean annual decrease of 2.1% per year, observed in the control treatment across sites. Error bars indicate the 95% confidence intervals. *Abbreviations: CT, control; SD, saw dust; MS, maize stover; TD, Tithonia; CC, Calliandra; FYM, farmyard manure.*





similar at all sites. The control treatment resulted in the largest decrease in SOC concentration, while the application of FYM at the rate of 4 t C ha$^{-1}$ yr$^{-1}$ led to the smallest decrease or even increase. The SD treatment generally showed similar decreases in SOC concentration as the control treatment, and CC and TD led to SOC change rates in between the control and FYM

treatments. The application of organic resources at the rate of 1.2 t C ha$^{-1}$ yr$^{-1}$ did, with a few exceptions (e.g., FYM in Embu and Sidada), not significantly reduce the decrease in SOC concentration compared to the control. Notably, the absolute annual SOC concentration decrease in the control treatment was largest in Embu (about 0.6-0.7 g C kg$^{-1}$ yr$^{-1}$), followed by Sidada (about 0.5 g C kg$^{-1}$ yr$^{-1}$), and was about 0.2 g C kg$^{-1}$ yr$^{-1}$ in Machanga and Aludeka. The picture was different in relative terms, with site specific annual decreases of about 2% to 3% of initial SOC in the control (Fig. 4). The highest relative SOC decrease

in the control treatments was observed in Machanga in the CT-N treatment, which lost almost 3% of initial SOC per year. Within control treatments across sites, significantly lower decreases than CT-N in Machanga were found in CT+N in Aludeka, CT-N in Embu and in both control treatments in Sidada, all losing about 2% per year.

In contrast to the control, the FYM treatment at 4 t C ha$^{-1}$ yr$^{-1}$ lost only about 0.4 g C kg$^{-1}$ yr$^{-1}$ (corresponding to 1% of initial SOC) in Embu, and about 0.1 g C kg$^{-1}$ yr$^{-1}$ in both Machanga (1% of initial SOC) and Sidada (0.3% of initial SOC), while in

Aludeka it gained about 0.1 g C kg$^{-1}$ yr$^{-1}$ (1.5% of initial SOC). In terms of relative SOC changes, differences between sites within the same organic resource treatment were observed more frequently than in the control. For example, the SOC gains in the FYM treatments at 4 t C ha$^{-1}$ yr$^{-1}$ in Aludeka differed significantly from the losses observed within the same treatments in Embu and Machanga. Also, the other 4 t C ha$^{-1}$ yr$^{-1}$ treatments showed in most cases significantly lower relative SOC decreases in Aludeka than in Machanga. Additionally, the +N treatments of CC and TD showed significantly lower relative

SOC decreases in Aludeka than in Embu and Sidada (Fig. 4).

In the site specific analysis, the addition of mineral N showed a significant interaction with the organic resource treatment, but, with a few exceptions, significant differences between +N and -N treatments were absent, nonetheless. Specifically, the CC treatment at 1.2 t C ha$^{-1}$ yr$^{-1}$ in Aludeka showed no significant SOC decrease in the +N treatment while the corresponding -N treatment lost about 0.1 g C kg$^{-1}$ yr$^{-1}$. Similarly, the +N subplots subject to 4 t C ha$^{-1}$ yr$^{-1}$ addition of either CC or TD in

Machanga, showed a significantly lower decrease in SOC concentration than their -N counterparts. Finally, the SD treatment at 1.2 t C ha$^{-1}$ yr$^{-1}$ in Embu, had a significantly lower decrease in SOC concentration in the -N treatment compared to the +N treatment, but for all other organic resource treatments at all sites, the +N and -N subplots were not significantly different from each other.

### 3.3 SOC stocks at different equivalent soil masses and the correspondence to SOC concentrations

Changes in SOC stocks based on equivalent soil masses are the most reliable way to assess carbon sequestration or losses, but in contrast to the SOC concentrations of our study, were only available for 2021. Hence, apart from the one time point SOC stock assessment (Fig. 5), we tested whether the 2021 SOC stocks for the top 2500 t ha$^{-1}$ equivalent soil mass were in alignment with the observed temporal trends of the SOC concentration in the 0 - 15 cm soil layer, i.e., whether soils with the highest losses in SOC concentration had the lowest SOC stocks in 2021. A highly significant correlation emerged between

these two types of SOC assessment in the planted plots, explaining 81% of total variation for the clayey sites, Embu and Sidada,



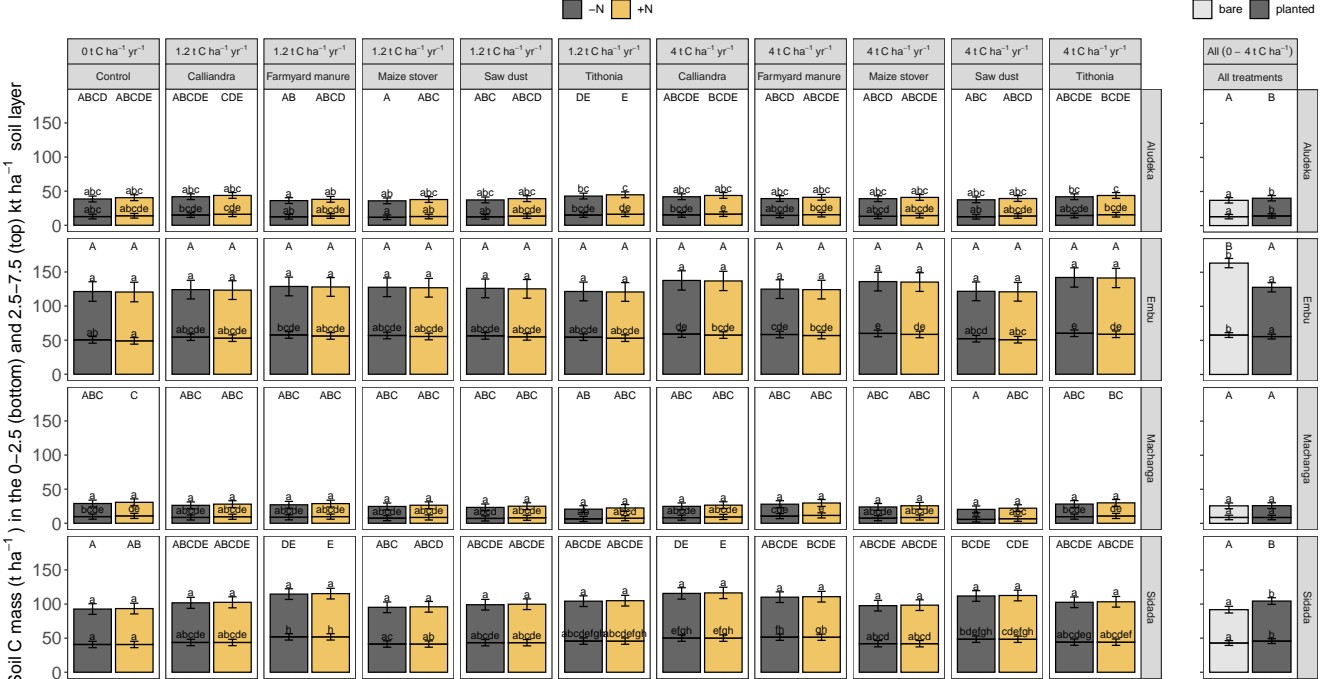

**Figure 5.** Estimated least square means for the SOC stocks across sites in the cumulative equivalent soil masses of 0 to 2500 and 2500 to 7500 t soil ha$^{-1}$ (bottom and top bar, respectively). As there was no significant interaction between the cropping status (being bare or planted) and the organic resource and N treatments, the results are displayed as the estimated least square means for different organic resource and N treatments for planted plots (left plot) and as mean SOC stocks from all organic resource and N treatments in the bare compared to planted plots (right plot). Error bars display the 95% confidence intervals. SOC stocks from the same site and soil mass layer which share no lowercase letter are significantly different from each other. Same capital letters indicate the absence of a significant difference between treatments for the whole 0 to 7500 t ha$^{-1}$ equivalent soil mass layer (p < 0.05).

and about 56% for the two coarse textured sites, Machanga and Aludeka (Fig. A2). Due to comprising of a one time output, less significant differences in SOC stocks were detected between treatments compared to SOC concentration. The 2021 SOC stocks also assessed bare plots, but there was no significant interaction between the plot status (being bare or planted) and the experimental treatments. Yet, the plot status had a site-specific effect on SOC stocks. In Aludeka and Sidada, significantly

higher SOC stocks were observed in the planted compared to the bare plots in both the top- and subsoil equivalent soil mass layers, while no effect was found in Machanga. Surprisingly, in Embu, the bare plots showed significantly higher SOC stocks than the planted plots in both soil layers (Fig. 5). Yet, subsoil SOC stocks in the 2500-7500 t ha$^{-1}$ soil layer showed no significant differences between treatments in any of the sites, except in Aludeka, where the TD treatments at both rates of C input had higher SOC stocks than the FYM treatment at 1.2 t C ha$^{-1}$ yr$^{-1}$. Besides, in contrast to the topsoil SOC stocks, the subsoil SOC

stocks were much less related to topsoil trends of SOC concentration, suggesting relatively small interactions between top- and subsoil SOC dynamics. In fact, a significant association between the temporal trends of topsoil SOC concentrations and subsoil





SOC stock was only found for two sites, i.e., in Machanga ($R^2$ of 0.24) and Sidada ($R^2$ of 0.45), yet much weaker increases of

SOC stocks were found for the subsoil compared to the topsoil (Fig. A2).

## 3.4 Apparent carbon storage efficiency as affected by site and organic resource type

**Table 3.** Estimated apparent CSE of SOC formation by treatment and site. Same lowercase letters at the same site indicate the absence of a significant difference between treatments at that site. Same uppercase letters of the same organic resource indicate the absence of a significant difference between sites within the same organic resource (all p < 0.05).

| Residue treatment | Site | apparent CSE (%) | 95% CI |
|---|---|---|---|
| CC [B] | Aludeka | 10[c] | 6 to 13 |
| FYM [A] | Aludeka | 13[c] | 10 to 17 |
| MS [A] | Aludeka | 3[ab] | 0 to 7 |
| SD [AB] | Aludeka | 1[a] | -2 to 5 |
| TD [B] | Aludeka | 9[bc] | 5 to 12 |
| CC [B] | Embu | 10[ab] | 6 to 13 |
| FYM [A] | Embu | 13[b] | 9 to 16 |
| MS [B] | Embu | 13[b] | 9 to 16 |
| SD [AB] | Embu | 4[a] | 1 to 8 |
| TD [B] | Embu | 11[b] | 8 to 15 |
| CC [A] | Machanga | 2[a] | -1 to 6 |
| FYM [A] | Machanga | 9[b] | 6 to 13 |
| MS [A] | Machanga | 1[a] | -2 to 5 |
| SD [A] | Machanga | -2[a] | -5 to 2 |
| TD [A] | Machanga | 2[a] | -1 to 6 |
| CC [B] | Sidada | 13[b] | 10 to 16 |
| FYM [B] | Sidada | 20[c] | 16 to 23 |
| MS [A] | Sidada | 3[a] | 0 to 7 |
| SD [B] | Sidada | 6[a] | 3 to 9 |
| TD [AB] | Sidada | 8[ab] | 4 to 11 |

The efficiency with which organic resources were converted into SOC varied by site and treatment (Table 3). The treatments with respectively the highest and lowest decrease in SOC concentration (FYM and SD), corresponded to the treatments with the highest and lowest $CSE_a$. The highest $CSE_a$ for FYM was found in Sidada (20%), while it was about 13% in Aludeka and Embu, and 9% in Machanga. On the other hand, the lowest $CSE_a$ for SD (-2%) was observed in Machanga, whilst it was between 1 and 6% in the other sites. Besides, the differentiation of FYM in terms of $CSE_a$ compared to other treatments varied among sites. In Sidada, the $CSE_a$ for FYM was significantly higher than in the other treatments except CC, in Embu it was only significantly higher than that of SD, and in Machanga and Aludeka, FYM had a significantly higher $CSE_a$ than MS and SD. On





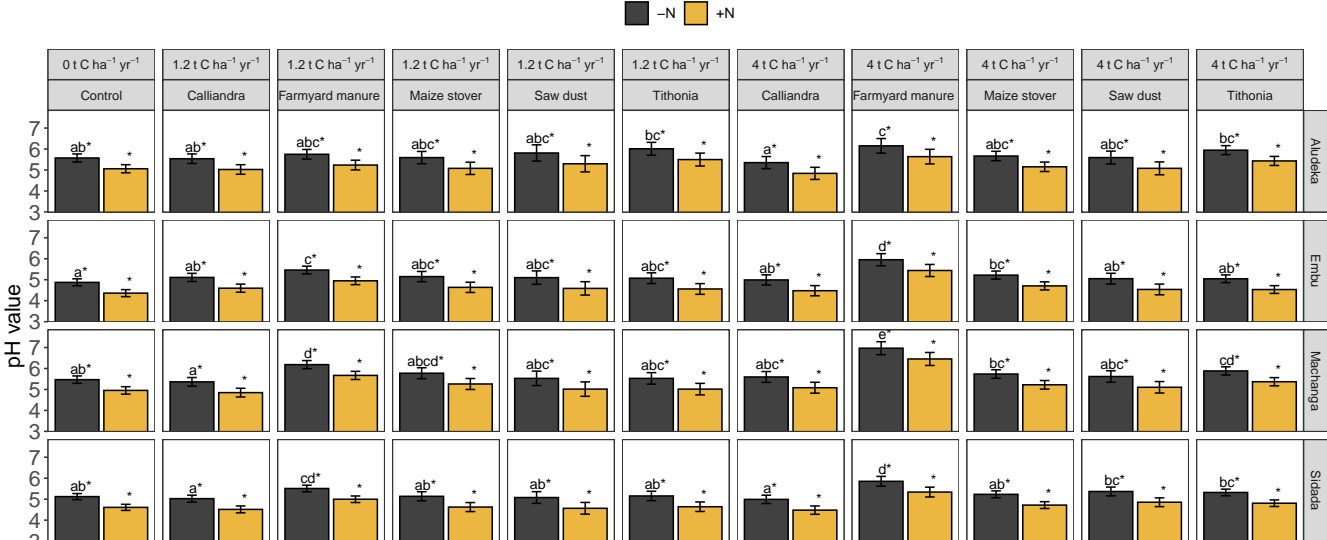

**Figure 6.** Estimated least square means for the topsoil pH value by site. Error bars display the upper half of the 95% confidence intervals. Organic resource treatments from the same site which share no lowercase letter are significantly different from each other ($p < 0.05$). The asterisk (*), indicates that a similar significant difference existed between the +N and -N treatment for all organic resource treatments and at all sites (due to the lack of a significant interaction of ±N with both organic resource treatments and site).

the other hand, the SD treatment was outperformed in terms of $CSE_a$ by the CC treatment in Aludeka and Sidada, and by the TD treatment in Aludeka and Embu. The MS treatment was in the group of lowest $CSE_a$ in all sites (1-3%) except in Embu, where, surprisingly, it had a $CSE_a$ of 13%, which was not significantly different from that for FYM. The $CSE_a$ within the same organic resource types further varied between sites, as indicated by a significant interaction of organic resource treatment with site. For example, FYM had a significantly higher $CSE_a$ in Sidada than in all other sites, while the $CSE_a$ of CC and TD was significantly lower in Machanga than in the other three sites.

### 3.5 The effect of organic resources and mineral N on soil pH

Mineral N application had a significant negative effect on soil pH, which was consistent across sites, i.e., the +N treatments had a pH of 0.11 units lower than the -N treatments. In addition, there was a highly significant interaction of organic resource treatment with site. In all sites, soil pH values were highest in the FYM treatment at 4 t C ha⁻¹ yr⁻¹, whereas the lowest pH values were found in the control or CC treatments. Yet, the strength of the differentiation of FYM at 4 t C ha⁻¹ yr⁻¹ compared to other treatments differed among sites. For example, the pH values in the FYM treatment at 4 t C ha⁻¹ yr⁻¹ in Embu (5.95 in -N) and Machanga (6.96 in -N) were significantly higher than in all other treatments, in Sidada (5.85 in -N) it was higher than all, except in the FYM treatment at 1.2 t C ha⁻¹ yr⁻¹, while in Aludeka (6.15 in -N), it was only significantly higher than that in the control and CC treatments. While the pH values in SD and CC at both C input rates were not significantly different from





that in the control in any site, TD at 4 t C ha$^{-1}$ yr$^{-1}$ had a significantly higher pH value than the control in Machanga, and a significantly higher pH value than those in CC at 4 t C ha$^{-1}$ yr$^{-1}$ in Aludeka and Sidada, as well as those in CC at 1.2 t C ha$^{-1}$ yr$^{-1}$ in Machanga and Sidada.

## 4    Discussion

### 4.1    In most sites, even high-quality organic resource addition could not completely halt SOC loss

The results of our study showed clearly that even at high rates of organic resource addition (4 t C ha$^{-1}$ yr$^{-1}$), SOC generally decreased (i.e., of four sites in this study, only Aludeka showed increased SOC). This was indicated by the changes in SOC concentrations and corroborated by their strong correlation with the 2021 topsoil equivalent soil mass based SOC stocks. Hence, very high amounts of high-quality organic resource inputs are needed to prevent a loss of SOC concentrations, with even 4 t C ha$^{-1}$ yr$^{-1}$ not being sufficient at Machanga and Embu. The high losses in Machanga, the site with the highest relative loss of almost 3% of initial SOC concentrations per year, are likely not only from SOC mineralization but also from erosion. It is, Machanga showed extremely strong signs of topsoil erosion, which was, however, not quantified. For Embu, the high initial SOC concentrations may be responsible that even the high loads of 4 t C ha$^{-1}$ yr$^{-1}$ of the farmyard manure treatment could not form enough new SOC to counterbalance the losses. Yet, even in Sidada, a very favorable site, SOC was lost in almost all treatments and none hat a significant gain. In conclusion, our first hypothesis, that high-quality organic resources increase SOC, is rejected for all sites except Aludeka.

The generally observed SOC losses in our study corroborate the results of Sommer et al. (2018), who reported similar SOC losses at two sites in western Kenya, both close to Sidada. Another recently published study from a long-term trial of compost application of about 3 t C ha$^{-1}$ a year in Ivory coast, could also not maintain initial SOC levels (Cardinael et al., 2022). It seems thus, that maintaining SOC in arable cropping in tropical systems is very difficult, even with high inputs that are aimed at replenishing SOC. Potential explanations for the high SOC losses in tropical soils are manifold. They have been attributed to humid climate (Ryan and Law, 2005; Todd-Brown et al., 2013), high weathering status of the soils and thus limited protection of SOC (Six et al., 2002; Doetterl et al., 2015), and high temperatures (Davidson and Janssens, 2006; Conant et al., 2011; Wei et al., 2014). If inconsistent but strong rains occur, as was the case in Machanga, erosion can also be considerable and wash away the SOC rich topsoil. Furthermore, a loss of SOC usually occurs when natural vegetation is converted to arable land (Sanderman et al., 2017) and about 50% of initial carbon is usually lost (Guo and Gifford, 2002; Lal, 2018). In the tropics, the SOC loss due to land use change is usually more severe than that and can be as high as 70-85%, and strongest in the initial few years (Solomon et al., 2007). Three sites of this study had been under shifting cultivation with considerable fallow time prior to the experiment establishment, while Machanga was even converted from natural vegetation. This suggests that soils were still in the initial rapid loss phase, as observed most strongly in the absolute decreases of SOC concentrations in Embu and Sidada, the sites with the highest initial concentrations. Hence, if starting at degraded lands with initially low SOC, it may be possible to increase SOC stocks with input of external organic resources (Sommer et al., 2018). The example of Machanga, indicates,



however that low initial SOC is not the only determinant for successful SOC increase and that suitable climate, as in Aludeka,
is also needed.

Depending on the initial levels, it is expected that a certain amount of SOC will initially be lost from soils that were only recently converted from native vegetation. The evidence of this study and others (Sommer et al., 2018; Cardinael et al., 2022) confirm this and indicate that tropical soils under maize monocropping will not be an absolute carbon sink in the sense that they still add $CO_2$ to the atmosphere even though they might be reducing emissions compared to the business as usual cropping
systems (i.e. the control treatment). Since it is unrealistic that with a growing population and living standard there is much land in SSA that will be spared from crop production, aiming to halt SOC loss at medium SOC levels may be the only possible scenario to maintain soil fertility. For example, in our experiments, several treatments that experienced a reduction in SOC concentration still experienced a gain in yield with time (Laub et al., ; in review). The results of our study demonstrate that the mitigation potential of improved management can still be substantial, i.e., between one and two thirds of the SOC losses in the
control could be avoided by farmyard manure additions of 1.2 and 4 t C ha$^{-1}$ yr$^{-1}$, respectively (Fig. 1). The farmyard manure treatments also had the highest yields (Laub et al., ;in review), suggesting an even better performance if emissions are yield scaled as they should be (Clark and Tilman, 2017). If however, the declared goal is an actual absolute removal of $CO_2$ from the atmosphere, one may need to target soils like in Aludeka, or include other agronomic measures, such as intercropping or agroforestry, that could add more C to the soil (Muchane et al., 2020).

**4.2 Organic resource quality has a strong effect on SOC buildup and loss**

Like several other studies (e.g. Galicia and García-Oliva, 2011; Laub et al., 2022; Li et al., 2022), our results corroborate the emerging paradigm that organic resources of high quality (i.e., with a low C/N ratio) are most effective in forming new SOC. This paradigm assumes that SOC is mostly of microbial origin (Denef et al., 2009; Cotrufo et al., 2013; Kallenbach et al., 2016) and that high-quality resources are processed with a higher carbon use efficiency (Manzoni et al., 2012; Sinsabaugh
et al., 2013). The results of our low-quality resources, such as saw dust and maize stover, which only marginally improved the SOC concentration over the control, are also in alignment with this paradigm. Similar results were reported in another long-term experiment in Thailand (Puttaso et al., 2013). The results of these studies combined suggest that abiotic condensation of humic substances from lignin-rich materials (Frimmel and Christman, 1988) does not play a major role in SOC formation, as was postulated in the 1980s (Woomer and Swift, 1994). This is highly relevant for ISFM, as earlier hypothesized trade-offs
between building SOC and providing plant nutrition by organic resources (Palm et al., 2001b) are in fact falsified by the results from long-term experiments. In contrast to these early hypotheses, organic resources that have a good synchrony of nutrient release with plant demand, such as farmyard manure and *Calliandra*, are also the most effective in SOC formation/maintenance. Together, these results show that the concept of ISFM remains highly relevant for soil fertility improvement.

Despite the trend of a general loss of SOC concentration in the topsoil layer (0-15 cm) across sites, the different rates of loss
for different organic resources show the importance of their quality in SOC maintenance. This was seen best at additions of 4 t C ha$^{-1}$ yr$^{-1}$, where farmyard manure was most efficient at all sites. In contrast, the low-quality saw dust could not reduce SOC losses compared to the control at any site. Our findings corroborated the results of several studies that demonstrated that





application of farmyard manure is the best option to sustain SOC (Sileshi et al., 2019; Rusinamhodzi et al., 2013; Mtangadura et al., 2017) and pH (Mucheru-Muna et al., 2014). Our study also confirmed that cut-and-carry green manures such as *Cal-*

*liandra* and *Tithonia* are effective in forming new SOC compared to no input (Kunlanit et al., 2014), yet with lower efficiency than farmyard manure. On the other hand, green manures were shown to be more effective in SOC maintenance compared to low-quality maize residues and sawdust, yet not as consistent across sites (e.g., not in Embu). The postulated advantage of class 2 organic resources, such as *Calliandra*, relative to class 1 organic resources, such as *Tithonia* (Kunlanit et al., 2014), was, however, not supported by our results. Kunlanit et al. (2014) hypothesized that a higher SOC formation of class 2 or-

ganic resources was related to the polyphenol and lignin content, increasing synchronisation between microbial demand and availability by preventing leaching of nutrients and dissolved organic carbon. A potential reason for similar performance of *Calliandra* and *Tithonia* could thus be, that they only differed in terms of polyphenol contents but not in lignin.

The question is, why farmyard manure, despite being similar to *Calliandra* in terms of C/N ratio and aromatic compounds (Table 2), was better in avoiding SOC losses. It could be that farmyard manure, which constitutes an already decomposed form

of biomass (e.g., about 30% of the original biomass intake by animals; Dickhoefer et al., 2021; Hossain, 2021), contains more stabilized forms of C that can, for example, directly attach to soil minerals (Angst et al., 2021). Alternatively, it is possible that the the high amount of lignin in combination with high nutrient concentration (i.e., the regulatory effects of N and aromatic components combined; Kunlanit et al., 2014) or the higher P concentration in farmyard manure (the only treatment with a C/P ratio < 100) made it most effective in terms of SOC formation. In the first case, lignin but not polyphenols would be

responsible for the regulatory effect, as lignin and polyphenols combined were similar in farmyard manure and *Calliandra*. Xiao et al. (2021) found that an increased microbial carbon use efficiency of farmyard manure was not only due to a favorable carbon-to-nutrient ratio, but also because its application increased soil pH. Mtangadura et al. (2017) in another long-term ISFM trial, also found that farmyard manure application significantly increased soil pH and SOC. Such a positive effect of farmyard manure on soil pH compared to the initial measurements was also found in all four of our experiments (Fig. 6) and it is possible

that the pH increase achieved by farmyard manure addition further alleviated constraints on carbon use efficiency introduced by low pH (Malik et al., 2018).

### 4.3 Effect of mineral N fertilizer

Contrasting our second hypothesis, low-quality saw dust or maize stover did not show lower SOC losses in the +N treatment than in the -N treatment (Fig. 4). The impossibility to enhance a poor organic resource stoichiometry by amending mineral

nutrients from external sources either indicates that organic resource quality is determined by more than just ratios of nutrients, or that it is difficult for microbes to counterbalance the poor quality of organic amendments by taking up nutrients in the mineral form. It is furthermore surprising that no differences in SOC loss between the +N and -N treatments of the control were found at any site. For example, Ladha et al. (2011) showed a global analysis that SOC does on average increase when mineral fertilizer is added, yet the effect from organic inputs was considerably stronger. A higher maize productivity for +N compared to -N

treatments was found in about half of the treatments of this experiment Laub et al., (in review), so the absence of ±N differences in SOC indicates that aboveground biomass productivity cannot automatically be assumed to translate into belowground inputs.

 

Plants may invest less into roots if they are supplied too well (Prescott et al., 2021). Additionally, Silva-Sánchez et al. (2019), studying forest soils, found a reduced microbial carbon efficiency in treatments of mineral N addition. This reduced efficiency may be related to root-exudate related priming (Kuzyakov et al., 2000), which could have been higher in the +N treatments.

Also, while the treatments in our study received basal P and K, it cannot be ruled out that this unresponsiveness to mineral N addition in the control was due to a lack of other essential nutrients (see e.g. Mtangadura et al., 2017). In this case, higher N availability might lead to higher priming as plants of higher biomass try to obtain other missing nutrients.

### 4.4 Organic resource additions reduce SOC loss, but further agronomic measures are needed to increase SOC

While in Sidada and Aludeka, SOC losses could be avoided with high inputs of 4 t C ha$^{-1}$ yr$^{-1}$ of farmyard manure, such

amounts are usually not achievable in smallholder systems in SSA (Rufino et al., 2006; Wawire et al., 2021). In fact, the limited availability of manure is one of the main factors creating the gradients of soil fertility and SOC on smallholder farms (Tittonell et al., 2013). The net primary biomass production in the study regions is around 6 t C ha$^{-1}$ yr$^{-1}$ (Running et al., 2004), so the high rates of 4 t C ha$^{-1}$ yr$^{-1}$ as manure are exceeding by far what is available at the landscape scale (Vanlauwe and Giller, 2006). For example, typical conversion rates of biomass C to manure C are around 30% (de Azevedo et al., 2021; Dickhoefer

et al., 2021), so even a ha of grassland can maximally produce manure equivalent to 2 t C per year. Increased organic resource inputs at one location will therefore happen at the cost of losses at another location, effectively only redistributing SOC in the landscape (Wiesmeier et al., 2020) and the typical gradients of lower soil fertility with increasing distance to homesteads (Vanlauwe et al., 2015; Kayani et al., 2021) already indicate that this redistribution is current reality.

This effectively means that to increase SOC at landscape scale, further agronomic practices that increase primary production

and biomass availability, such as intercropping, rotations with grasses or agroforestry practices (Corbeels et al., 2019; Tessema et al., 2020), are needed. Additionally, the focus should be on direct plant inputs from the roots, which is considered the most efficient pathway of SOC buildup (Prescott et al., 2021; Sokol et al., 2019). Both roots and root exudates are known to form new SOC with higher efficiency than external organic resource inputs (Rasse et al., 2005; Jackson et al., 2017; Sokol and Bradford, 2019), partly due to the fact that microbes can easily assimilate root exudates, which contributes to the formation

and stabilization of microbial necromass (e.g. Wang et al., 2022), partly due to the proximity of inputs and microbes. The latter has been highlighted as the most important factor (Lavallee et al., 2018), so focusing only on the quantity of C inputs and ignoring quality may be misleading, as demonstrated by the poor performance of adding 4 t C ha$^{-1}$ yr$^{-1}$ saw dust and maize straw in this study (Fig. 1).

In the light of the results of our study, suitable measures should thus increase the input quantity and quality at the same

time. For example, intercropping can produce more biomass than sole crops on the same surface due to complementary use of resources (e.g., light, nutrients, water, etc.; Malézieux et al., 2009; Bedoussac et al., 2015). Especially the high-quality inputs from cereal-legume intercropping were shown enhance SOC build up in the long term compared to sole crops (Li et al., 2021b). If intercropping is not suitable, crop rotations with legumes (Cong et al., 2015) and genotypes with strong root systems (e.g. Van de Broek et al., 2020), may be alternatives. Our data partly shows the importance of roots by the higher SOC stocks in

planted compared to bare plots at both equivalent soil mass layers in Aludeka and Sidada (Fig. 5) - both of these sites had the





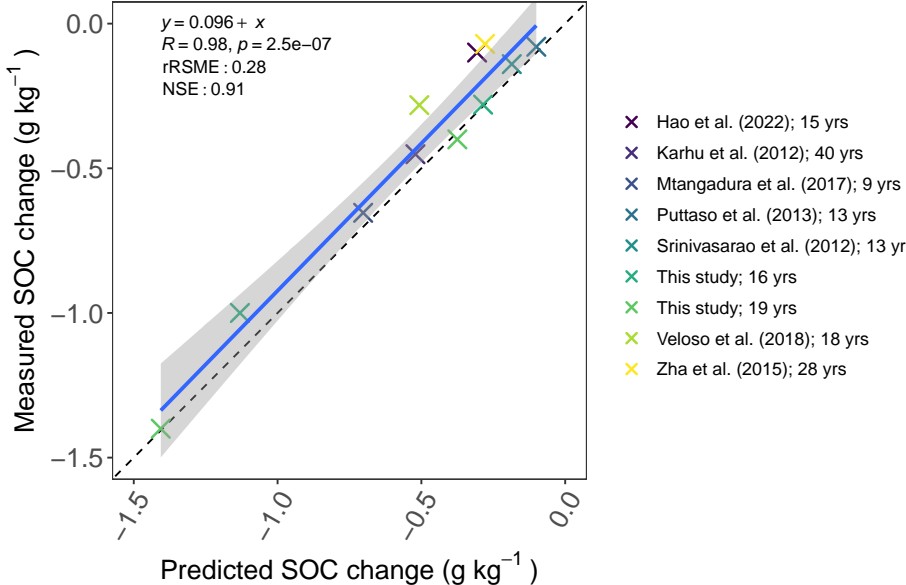

**Figure 7.** Comparison of predicted vs measured SOC change at the end of different experiments without input of external organic matter and fertilizers assuming a 1.5% per year first order decomposition at 10°C mean annual temperature. The decomposition rate was scaled with an exponential temperature function with a $Q_{10}$ of 2 using the mean annual temperature of each site. The legend shows the different experiments (Srinivasarao et al., 2012; Karhu et al., 2012; Puttaso et al., 2013; Zha et al., 2015; Mtangadura et al., 2017; Veloso et al., 2018; Hao et al., 2022), and their duration in years (yrs). *Abbreviations: rRSME, relative root mean square error; NSE, Nash-Sutcliffe model efficiency.*

highest maize biomass of all four sites (data not shown, here; Laub et al., ;in review). However, the contrasting results from Embu, with higher SOC stocks in bare compared to planted plots do not fully align with this. We cannot exclude that this was an artifact of the different and partially open soil auger we had to use in Embu for the 2021 sample - the other auger type repeatedly broke due to the very hard subsoil. In any case, the strong losses of SOC in the control with and without mineral

N at all sites indicate that despite the presence of root inputs, maize monocropping inputs cannot counterbalance SOC loss. This is in alignment with estimates of less than 1 t C ha$^{-1}$ and year of root inputs and rhizodeposits combined for a similar experiment (Cardinael et al., 2022).

## 4.5 Site specificity in the formation of new SOC

An interesting pattern of site specificity emerged from our data. The decrease in SOC concentration in the control was rather

uniform, e.g., about 2% of initial SOC per year, with Machanga being one exception as a result of erosion losses (-N treatment; 3%). This seems not to be limited to our sites, because comparing our data with published data of SOC loss from no-input experiments around the globe showed a fairly consistent trend of SOC loss across studies (Fig. 7). In contrast to this rather uniform loss, the response of SOC concentrations to high-quality organic resource additions were more site specific. This was



indicated by large differences in SOC formation in the farmyard manure, *Tithonia* and *Calliandra* treatments at 4 t C ha$^{-1}$ yr$^{-1}$

between sites (Fig. 4). Also, the $CSE_a$ of all organic resources was site-specific (i.e., significantly lower in Machanga compared to other sites, and in the case of farmyard manure, higher in Sidada compared to the other sites, (Table 3). This cannot be due to texture alone: Aludeka and Machanga have a similar texture but a significantly higher $CSE_a$ was found for *Calliandra* and *Tithonia* in Aludeka compared to Machanga. Also, Sidada and Embu are similar in texture, but $CSE_a$ of farmyard manure was significantly higher in Sidada than in Embu. Thus, we reject our third hypothesis that organic resource quality is of higher

importance for SOC formation than soil properties, as the difference between sites in $CSE_a$ was as large as between organic resources.

Possibly, there are differences in mineralogy between sites, which, under similar inputs, are the main explanatory factor for differences in SOC stocks (Zech et al., 1997; Reichenbach et al., 2021; Keller et al., 2022), next to texture (Mainka et al., 2022). For example, mafic parent material has shown a higher potential to store SOC in aggregates than felsic material

(Reichenbach et al., 2021). As a consequence, it is unlikely that rainfall and temperature combined with texture are sufficient to explain differences in $CSE_a$, as put forward by some authors (Schimel et al., 1994; Smith and Waring, 2019). Even a temperature dependent $CSE_a$ (Frey et al., 2013) would not suffice to explain the higher $CSE_a$ observed in Aludeka compared to Machanga (*Calliandra*, *Tithonia*) or in Sidada compared to Embu (farmyard manure): Sidada has higher temperatures than Embu and in Aludeka and Machanga temperatures are similar. Clearly, a better understanding of the interactions of organic

resource quality, climate and soil mineralogy are needed to effectively target soils that have the potential to sequester SOC. This calls for standardized manipulation experiments with a range of different quality inputs across a range of mineralogies and climates. Standardized meta-analyses could present suitable alternatives, but their interpretability usually suffers from experimental designs being dissimilar. Only an improved understanding of all factors influencing $CSE_a$ can help to satisfy the local adaptation criterion of ISFM (Vanlauwe et al., 2015) so that smallholder farmers can apply organic resources and mineral

fertilizer where they are most effective.

## 5   Conclusions

This study showed that continuous maize cropping in sub-Saharan Africa without organic resource inputs are subject to a significant loss of on average 2% per year of their initial SOC concentrations (ca. 40% over 19 years). Site specificity exited: While the addition of 4 t C ha$^{-1}$ yr$^{-1}$ of high-quality organic resources could counteract the losses in all four sites, a complete

halt of SOC loss or even achieving gains, was only possible at two sites. Farmyard manure application was the most effective treatment, but 4 t C ha$^{-1}$ yr$^{-1}$ of green manures from *Calliandra* and *Tithonia* still had a significantly lower SOC concentration losses than the control. With the exception of maize stover at one site, the application of low-quality organic resources such as saw dust and maize stover did not help to reduce SOC losses, even at rates of 4 t C ha$^{-1}$ yr$^{-1}$. Mineral N application did also not help much to reduce SOC losses, and it did not improve carbon stabilization from low-quality organic resources.

Differences between sites in the efficiency of high-quality organic resources to form new SOC were found to be stronger than differences in loss of SOC in the control, which showed the importance of soil properties in the effectiveness of ISFM. Despite



site specificity in the efficiency to from new SOC, our results showed clearly, that farmyard manure application was the most effective treatment for SOC formation at all sites. The application of farmyard manure, which is realistically only possible at rates around 1.2 t C ha$^{-1}$ yr$^{-1}$, should thus be a priority to maintain SOC stocks and soil fertility in tropical soils. In addition 490 crops that increase carbon inputs through roots may be needed if a complete halt of SOC loss is the aim.

*Data availability.* The dataset used for this study, including SOC and N is made available under the IITA data repository https://doi.org/10. 25502/wdh5-6c13/d

*Author contributions.* BV and JS designed the research. MWMM, DM, SMN and WW managed and maintained the long-term trials. ML, AC, SMN, MN, WW, MvdB and JS were involved in the various sampling campaigns. MC, MN, BV and JS acquired funding for the research. 495 ML summarized the data, did the statistical analysis and prepared the original draft. All coauthors contributed in writing and editing of the final submitted article.

*Competing interests.* Johan Six is an executive editor of SOIL. All the other authors declare that they have no conflict of interest.

*Acknowledgements.* We want to thank Silas Kiragu, who is responsible for maintaining the trials in Embu and Machanga site, and John Mukalama, who implemented and maintained the trials in Aludeka and Sidada. Also, we want to thank John Waruingi for helping with 500 sample processing over the years, and Dr. Moses Thuita for coordinating the trials for many years, and Britta Jahn-Humphrey for organizing and overseeing the measurement of most SOC and TN in recent years. Additional a thanks to Moritz Bach and Julian Koller for their contribution in the measurement of SOC and TN from 2021 and to Lukas Burgdorfer and Franziska Büeler for their contribution to the data from 2018. This study was funded by the Swiss National Science Foundation (SNSF; grant number 172940) and by the DSCATT project "Agricultural Intensification and Dynamics of Soil Carbon Sequestration in Tropical and Temperate Farming Systems" (N° AF 1802-001, 505 N° FT C002181), supported by the Agropolis Foundation ("Programme d'Investissement d'Avenir" Labex Agro, ANR-10-LABX-0001-01) and by the TOTAL Foundation within a patronage agreement. We further acknowledge funding and technical support from the Tropical Soil Biology and Fertility Institute of CIAT (TSBF-CIAT) and the International Institute of Tropical Agriculture (ITTA) in maintaining the experiments throughout many years.



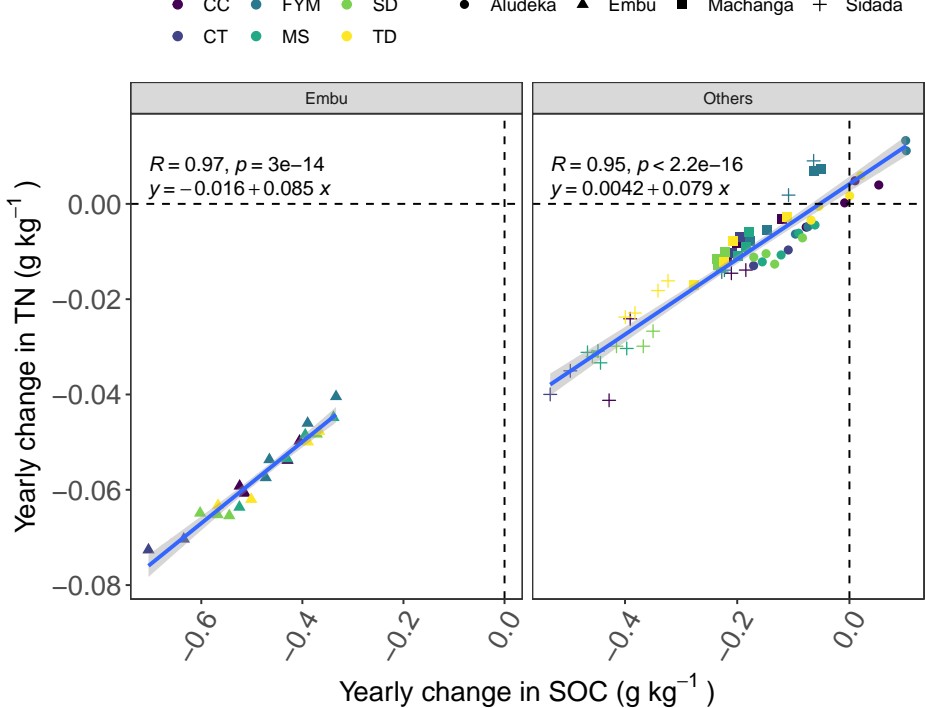

**Figure A1.** Correlation of temporal trends of SOC and TN. Displayed are the site specific least square means for both.





**Figure A2.** Estimated least square means of the SOC stocks in the cumulative equivalent soil masses 0 to 2500, 0 to 7500 and 2500 to 7500 t ha$^{-1}$ soil mass (top, middle and bottom graph, respectively) plotted against the least square means for the temporal trend of SOC concentrations in the 0-15cm topsoil horizon



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
