# Peer review of "Managing soil organic carbon in tropical agroecosystems: Evidence from four long-term experiments in Kenya"

_EGUsphere, 2022_

## Author Response (AR1)

**RC1:**

**General comments**

This manuscript addresses the question whether organic resource quality, quantity, mineral fertilizer or site properties are most important in counteracting SOC loss under continuous maize cropping in central and western Kenya. The topic is relevant to a larger community of readers because it shows that the application of organic and mineral fertilizers cannot completely counteract SOC losses across sites of different soil properties. Based on repeated measurements over time (2002 to 2021) using a split plot design, the authors build mixed linear models to show that the reduction of SOC concentration during 19 years ranged from 42 % to 13 % in spite of adding organic and mineral fertilizer. The authors conclude that a complete halt of SOC loss is not possible even with applications of 4 t C ha-1 yr-1. However, on the landscape-scale only rates of 1.2 t C ha-1 yr-1 are realistic without risking losses of SOC and soil fertility at other locations. This shows that on deeply weathered soils more site-specific measurements are needed beyond the application of organic and mineral fertilizer to maintain SOC. In particular, due to the lack of existing long-term studies on the behavior of tropical soils, there would be an added value to the study. In general, the manuscript is well written and the data supports the conclusions. There are some aspects in the methods section, which needs to be addressed to enhance clarity. In addition, there are parts in the discussion section that are not necessarily needed and detracting from the storyline of the manuscript. Please see my comments on that below. If these concerns can be addressed, this would be a suitable paper for EGUsphere.

Again, we want to express our sincere thanks for taking the time to give us this detailed and constructive feedback. We agree that the points are all valid and we are confident that addressing them has improved the manuscript considerably. We have put special focus on the main points raised 1) clarifying definitions and methodology (CSE and statistical model) 2) Improving figure design and captions 3) shortening and streamlining the results and discussion, and removing parts that detract from the main message of the manuscript (pH, C-sequestration discussion, to much CSE discussion), or putting them in the annex/supplement 4) moving ill-placed parts to the methods (Fig. 7, potential SOC stocks pitfall Embu). A detailed response to each individual comment and what we have changed as a result in the review phase of the article is attached below (our comments in blue).

**Below are the edits and comments on the manuscript**

**Abstract**

Well written and very interesting discussion especially in line 23 – 25.

The study sites have been management for at least 16 years but there was a recent conversion from permanent vegetation to agriculture. Does this mean the permanent vegetation was already managed before the land conversion?

**Yes, but the exact duration is not clear. We clarified this statement.**

N fertilizer or no fertilizer was the split-plot treatment. At all four sites, a loss of SOC , rather than gain, was predominantly observeddue to a recent conversion from permanent vegetation to agriculture. The average reduction of SOC concentration,

5 likely because the sites had been converted to cropland only a few decades before the start of the experiments. Across sites,

**How representative is Maize cropping for the study region?**

**We added a note on this in the first sentence.**

**прыаль**

In sub-Saharan Africa, maize is one of the most important staple crops, but long-term maize cropping with low external

**Introduction**

Overall well-structured and well written.

Line 43: Stabilization capability of SOC or what? I suppose you are talking about the reactivity of mineral surfaces towards sorption of organic matter in this context. You need to be clear here and elsewhere in the manuscript.

**Yes, thanks. We did so.**

to be a result of initially high SOC levels at the sites, favorable conditions for SOC decomposition, and a reduced stabilization capability of the 1:1 kaolinite clay minerals in tropical soils for SOC, due to low surface reactivity (Six et al., 2002; Sommer

Line 46 – 48: I would leave this part out since the focus of this paper is the interaction between fertilizer application and site properties and not the impact of land use history.

**We agree.**

adaptation to different soil conditions is needed. Land eover history also explains site specific SOC dynamics and can help to define the local potential for SOC sequestration. Indeed soils cultivated as pasture or forest have initially high SOC stocks that are difficult to maintain under cultivation (Lal, 2004).

**Line 72: What do you mean with SOC dynamics in this context?**

**We clarified this.**

As the main long-term goal of ISFM is to increase short-and short- and long-term soil fertility, in particular through increasing SOC, there is a need to better understand the effect of extent to which the rate and quality of organic resource additions on SOC dynamics influence the rate of SOC change under different pedo-climatic conditions. This can help to an-

Line 76 – 77: This belongs to the method section.

Agreed, we changed the sentences in a way that is more in line with an introduction.

and mineral N fertilizer affects SOC dynamics. Therefore To shed light on these questions, we analyzed data from four long-

3

term experiments conducted at four different sites in central and western Kenya(established in 2002 and 2005, respectively). All four experiments had the same treatments with organic resource additions of the same quality, with exactly the same organic and mineral resource additions at each site. The aims were to study objectives of the study were (i) to quantify the

Line 88 – 89: Why would the resource quality be of more importance than site conditions? What do you exactly mean with site conditions? Does it refer to soil mineralogy, climate, or both?

**We reformulated this.**

 The efficiency with which new SOC is formed is influenced by both the site-pedo-climatic conditions and the organic resource quality, but the resource quality plays the strongest role. Hence, we expect a significant interaction of resource quality with site in the efficiency to store new SOC.

**Material and methods**

A general question. How did you account for the changing soil bulk density (due to land conversion and land management) over time when calculation SOC stocks?

We did not have enough data to do such a correction. Hence, our CSE calculations are only an approximation, which we now clarified in the methods.

Line 103 – 107: Are the soil classifications based on lab data from the reference soil profiles?

Yes, from the time of establishment. We added this info.

Line 115 – 121: I suggest providing a figure here visualizing your plot and sampling design. This would also help to understand the structure of the random fixed models better. A table showing the sampling dates for the different study sites and what was sampled (topsoil, subsoil, BD etc.) would be of added value.

Thanks for this suggestion. We added these to the supplement.

Figure A1. Correlation Example of temporal trends the split plot design of SOC and TNthe long-term trials: Aludeka. Displayed are Red areas indicate the site specific least square means for bothbare fallow plot.

| Sampling dates | Sites sampled                   | Properties samples | Depth          |
|----------------|---------------------------------|--------------------|----------------|
| 2002           | Embu, Machanga                  | SOC, BD            | 0-15 cm |
| 2004           | Embu, Machanga                  | SOC, BD            | 0-15 cm |
| 2005           | Embu, Machanga                  | SOC, BD            | 0-15 cm |
| 2005           | Aludeka, Sidada                 | SOC                | 0-15 cm |
| 2006           | Embu, Machanga                  | SOC                | 0-15 cm |
| 2008           | Embu, Machanga                  | SOC                | 0-15 cm |
| 2012           | Embu                            | SOC                | 0-15 cm |
| 2013           | Embu, Machanga                  | SOC                | 0-15 cm |
| 2015           | Embu, Machanga                  | SOC                | 0-15 cm |
| 2018           | Embu, Machanga, Aludeka, Sidada | SOC                | 0-15 cm |
| 2019           | Embu, Machanga, Aludeka, Sidada | SOC                | 0-15 cm |
| 2021           | Embu, Machanga, Aludeka, Sidada | SOC, BD            | 0-15-30-50 cm  |

Table A1. Overview of the soil data that were available for this study.

**Did the bare and control plot received N, P and K fertilizer?**

**Yes, we clarified this.**

roots (and root exudates). In addition, a randomly allocated quarter of each split plot was kept as bare fallow throughout the entire duration of the experiments, receiving the exact same inputs but with no maize planted and with all emerging weeds removed by regular weeding. This was done to study the SOC dynamics without any additional inputs from roots or

Line 146: How exactly was the soil moisture content measured?

We added this information. Please see below.

Line 145: Was the soil bulk density measured before or after the 8 mm sieving?

**We agree that we had not described this part well, and it is clarified now.**

storage until further analysis in the laboratory. Samples were then For further analysis, a sub-sample was broken and crushed
 by pestle and mortar and sieved through a 2 mm sieve. Prior to Stone contents of both sieving steps were recorded. The soil moisture content of air-dried samples was based on drying a sub-sample at 105°C for 24h and subsequently calculating the dry soil weight per known core volume. Bulk density was determined based on the calculated absolute dry weight of stone-free samples and and the stone-corrected volume of soil cores, assuming a density of stones of 2.65 g cm-3. At all sites but Aludeka (with 1, 5 and 12% of stones in 0-15, 15-30 and 30-50 cm depth, respectively), stone content was very low (below 1% on

175 average). Prior to C and N analysis, samples were finely ground with a ball mill, then soil C and N concentrations contents

Line 150: Was the soil pH measured on the 2 mm fraction or on powdered samples?

**On the 2mm fraction. This info was added.**

were measured by dry combustion using an elemental analyzer (CHN628, LECO Corporation, Michigan, USA). In addition, soil pH (H2O) was determined on the unmilled 0-15 cm topsoil samples (2-mm sieved) taken in 2018. Because soil pH values

Line 152: Did you also tested the subsoil samples from the latest sampling campaign for carbonates? Did you tested with HCl because subsoil samples were not analyzed for soil pH? Please correct me if I´m wrong.

We did not test for carbonates but know from earlier samplings that pH < 6.5 until at least 60 cm depth and therefore the presence of carbonates is impossible. We added this info.

contents. From additional deeper soil sampling down to 60 cm depth carried out in 2018 (data not shown here), it is known that
the soil pH was lower than 6.5 in all depths considered.

Line 177 – 180: Could you please explain this step one more time for me?

We agree that this was a bit hard to understand have not detailed the individual effects. We hope that it is now more clear.

experiment (as initial measurements were not available for all plots). The initial fixed effects in the model were interactions random-site model were 1) time since the experiment started, 2) the interaction of time since the experiment started with the organic resource treatment, the 3) the interaction of time since the experiment started with the mineral N treatment and the

210 interaction of organic resource treatment with mineral N treatment() the three-way interaction of time since the experiment started with organic resource and mineral N treatments. In the site specific model, all these-fixed-site model, the fixed effects were further allowed to be site specific, by adding an interaction with site for each of them. In this model, site was the only fixed effect that was also allowed to have an intercept of its own and not only an interaction with time. 1) the interaction

8

of site with time since the experiment started, 2) the three-way interaction of site with time since the experiment started and with the organic resource treatment, 3) the three-way interaction of site with time since the experiment started and with the mineral N treatment and 4) the four-way interaction of site with time since the experiment started, organic resource and mineral N treatments. Moreover, the fixed-site model, had a fixed site effect without interactions. This assured that different

Line 186: Can you explain me what site-specific variance means in this statistical context?

We allowed a site-specific residual variance. This has been specified.

225 sampling in different years. Additionally, visual inspection of model residuals revealed variance heterogeneity between sites. Consequently, a site dependent residual variance was allowed in the models. After selecting an appropriate random effects

Line 202 – 205: It is hard to follow how you calculated the carbon storage efficiency. Could you please explain it to me again? I think this is an important part, which needs to be clear and easy to follow.

We agree that this important part was not clearly enough explained. We rewrote this section to more clearly describe how we transformed trends in SOC (%) to SOC stock estimates, how we fit the linear regression of SOC stock trends to annual C additions and how we interpreted the slopes of these regressions as CSE.

**2.4.2 Estimation of carbon storage efficiency**

From the temporal trends of SOC eoncentration content, we further estimated the change of SOC stocks in 0-15 em depth, with

- 240 the goal to derive the derived an estimate of the apparent carbon storage efficiency (CSEa) of the different organic resources in the 0-15 cm soil layer, i.e., a measure of efficiency to retain C. The CSEa has been defined as the fraction of C inputs contributing to C storage in the soil (Manzoni et al., 2018), e.g., in our case how much the annually added C through organic resources is found in the soil changed the trend of SOC stocks compared to the control treatment. To do so, we multiplied first the least square means of the change in annual SOC concentration content obtained by site and treatment from the mixed
- 245 model, with the mean-fixed-site model had to be transformed to mean annual change in SOC stocks. The mean BD of each site estimated from topsoil measurements to obtain the mean annual change in SOC stocks was used for this (treatment-specific differences in BD were absent). These BD estimations were also derived using a mixed linear model, from the available BD

**9**

measurements which had been conducted in the experimental year 1, 2 and in the calendar year 2021 at each of the sites. Then,
a site- and organic resource-specific linear regression between the estimated. This model did not contain any temporal trend;
different methods of BD measurements in initial years and in 2021 were used (core method vs direct measurement from the soil augers), so any trend would have likely been an artefact. While assuming this constant BD for SOC stock calculations is only a rough approximation and not fully consistent with the equivalent soil mass approach, we nonetheless considered that it was a valuable approach to quantify CSEa. Another potential limitation of these calculations considering only the 0-15 cm soil

layer (soil layer in which organic resources were incorporated), is that CSEa may be underestimated if significant portions of
 carbon inputs are stabilized in deeper soil layers from e.g.leached dissolved organic carbon.
 In the final step of CSEa estimations, a linear regression was fit, with the calculated mean annual change in SOC stocks for

0-15 cm as the response variable and the amount of annual organic resource C applied was fit, using a site-specific intercept. C input as the independent variable (i.e., 0, 1.2 and 4 t C ha-1 yr-1). These regressions were site- and organic resource-specific, so that estimates of CSEa per site and organic resources could be compared:

(1)

**$260 \quad dSOC = Site + C_{in} : OR + C_{in} : OR : Site$**

Here, *dSOC* is the mean annual change in SOC stocks in 0-15 cm (t C ha-1 yr-1). Site is the site specific intercept, Cin, the amount of annual C input (t C ha-1 yr-1) and OR the type of organic resources. Note that ":" represents interactions and that there was no OR-specific intercept. The intercept was set to site specific, i.e., not allowed to vary between different organic resources at the same site (i.e., the SOC change at 0 t C ha-1 yr-1 did not vary between treatments). The slope of this regression was taken on the numerical variable Cin represented the yearly change in SOC stocks (in 0-15 cm) per t C ha-1 yr-1 of organic resources added. It was thus interpreted as an estimate for CSEa of the different organic resources at the different sites (Manzoni et al., 2018) and we tested whether significant differences existed in the CSEa between the between slopes for different organic resources at the and at different sites (i.e., testing for a significant effect of organic resource treatment, site and their interaction). Estimated least-square means of the slope were converted into percent from t C t C-1 by multiplying them by 100.

**Results**

Figure 1: For me this is the key figure of the manuscript thus I have some questions for clarification. Is this the data normalized to the initial SOC content? Are the bars showing the data from all years/ sampling campaigns? What is the sample size for each bar? I would add that information in the caption.

Thanks for pointing this out. We added all this requested information to the caption.

Figure 1. Annual changes of SOC concentrations-contents in the top 0-15 cm soil layer in different organic resource treatments across sites. Annual The annual change in SOC concentrations is contents represent the slope of the regression against experimental time from the random-site model, using all available data (i.e., 5 (Sidada, Aludeka) or 14 (Embu, Machanga) repeated measurements from 3 replicates at 4 sites, comprising both mineral N treatments due to absence of mineral N effect). They are displayed as percentage of initial SOC (intercept of the model set to 100%). Treatments that share with the same capital letters-letter do not differ significantly from each other in the annual change of SOC (all p